# A higher proportion of ermin-immunopositive oligodendrocytes in areas of remyelination

**Intakhar Ahmad[1,2], Stig Wergeland[2], Eystein Oveland[3], Lars Bø[1,2]***

**1** Department of Clinical Medicine, University of Bergen, Bergen, Norway, **2** Department of Neurology, Norwegian Multiple Sclerosis Competence Centre, Haukeland University Hospital, Bergen, Norway, **3** Department of Biomedicine, Proteomics Unit at the University of Bergen (PROBE), University of Bergen, Bergen, Norway

* lars.bo@helse-bergen.no

**Data Availability Statement:** The authors of the study have not received any additional clinical information on the deceased for the Norwegian MS Registry and and Tissue Bank, and do thus not have any additional medical information available.

## Abstract

Incomplete remyelination is frequent in multiple sclerosis (MS)-lesions, but there is no established marker for recent remyelination. We investigated the role of the oligodendrocyte/myelin protein ermin in de- and remyelination in the cuprizone (CPZ) mouse model, and in MS. The density of ermin+ oligodendrocytes in the brain was significantly decreased after one week of CPZ exposure (p < 0.02). The relative proportion of ermin+ cells compared to cells positive for the late-stage oligodendrocyte marker Nogo-A increased at the onset of remyelination in the corpus callosum (p < 0.02). The density of ermin-positive cells increased in the corpus callosum during the CPZ-phase of extensive remyelination (p < 0.0001). In MS, the density of ermin+ cells was higher in remyelinated lesion areas compared to non-remyelinated areas both in white- (p < 0.0001) and grey matter (p < 0.0001) and compared to normal-appearing white matter (p < 0.001). Ermin immunopositive cells in MS-lesions were not immunopositive for the early-stage oligodendrocyte markers O4 and O1, but a subpopulation was immunopositive for Nogo-A. The data suggest a relatively higher proportion of ermin immunopositivity in oligodendrocytes compared to Nogo-A indicates recent or ongoing remyelination.

## Introduction

Multiple sclerosis (MS) is a chronic inflammatory demyelinating disease of the central nervous system (CNS) [1] and is a common cause of acquired neurologic disability in young adults [2]. In MS, there are multifocal areas, lesions of demyelination, inflammation, blood-brain-barrier breakdown, axonal and oligodendrocyte loss in the CNS [3]. There is frequent remyelination in MS lesions, starting at the early stages of lesion formation. Remyelination may include the recruitment of oligodendrocyte precursor cells (OPCs) and their differentiation into mature oligodendrocytes, which may extend processes to and enwrap demyelinated axons. The spirally enwrapped processes are then compacted to myelin. Remyelinated parts of an MS lesion are identified microscopically as areas with myelin staining being thinner and less organized, having uniformly thin myelin sheaths compared to the axon diameter [4, 5]. Typically, the

Raw cell count data are supplied in an supplementary data file.

**Funding:** The authors received no specific funding for this work.

**Competing interests:** The authors have declared that no competing interests exist.

remyelinated areas are found at the edge of a lesion, and the majority of lesions are incompletely remyelinated. The proportion of new MS-lesions in which remyelination occurs decreases over time [6].

Enhancing remyelination is an objective of many MS treatment trials, as this may confer neuroprotection, and less chronic disease progression [7]. There is a need for identifying sensitive and specific laboratory and radiological biomarkers for remyelination, both for use in clinical trials, and for a better understanding of the mechanisms of remyelination in MS. Currently, a reliable clinical or paraclinical biomarker for remyelination is not available [8].

In the cuprizone mouse model, oral administration of cuprizone (CPZ) continuously for five weeks results in oligodendrocyte loss, microglial activation, extensive demyelination and gliosis [9]. Remyelination occurs both during and after ending CPZ exposure, resulting in almost complete remyelination a few weeks after cuprizone exposure is ended [10, 11]. An increase in density of oligodendrocyte precursor cells is evident one week after starting CPZ exposure, and remyelination is detectable as early as after three weeks of ongoing CPZ exposure [10]. The CPZ model is considered appropriate for studying the remyelination process in MS, since several pathophysiological aspects of de- and remyelination are similar [12].

Ermin, also known as juxtanodin, is an oligodendrocyte-specific protein, known to be concentrated at the tip of F-actin-rich processes in the cytoskeleton synthesis [13–17]. Ectopic expression of ermin induced the formation of oligodendrocyte protrusions [13]. Ermin has been shown to be localized to the outer cytoplasmic lip of the myelin sheath, and the paranodal loops [13–16]. Ermin is a minor myelin protein, representing less than 1% of the total protein content [18]. Ermin knockout mice exhibits an aberrant myelin architecture in aged mice, with the splitting of myelin layers and display an increased sensitivity to cuprizone demyelination [19]. It is suggested that ermin may play a crucial role in myelin formation, in wrapping of axons and/or myelin compaction [13].

The objective of this study was to study whether the relative density of ermin-immunopositivie oligodendrocytes may be a marker of ongoing remyelination. We studied the extent and distribution of ermin immunopositivity during de-and remyelination in the cuprizone model and corroborated the findings by studying de- and remyelinated areas in MS-brain. The results indicate that the relative proportion of ermin-expressing oligodendrocytes, compared to the mature oligodendrocyte marker Nogo-A, may serve as a marker of remyelination in the CNS.

## Materials and methods

The study was approved by the Regional committee for medical research ethics of Western Norway (permit ID: 2013–560). The autopsy tissue was obtained from the National MS biobank & registry, the use of tissue for MS-research has been approved by the Regional committee for medical research ethics for Western Norway (Permit No. 046.03) The Norwegian Animal Research Authority approved the protocol. Consent to the use of tissue for research had earlier been given by the deceased, or in writing by their next of kin. It then had to have been made likely that this was in line with the wish of the deceased. The patients and/or the next of kin to the patients included had provided written consent to have data from the patients medical records used in research. The medical records of the patients included were anonymized

### Mice

Female c57Bl/6 mice (total n = 54) were acquired from Taconic (Ejby, Denmark) at the age of 7 weeks. The mean weight was 18.5 g (range 15.9–21.1). The mice were caged in Macrolon IVC-II cages (Scanbur, Karlslunde) in standard laboratory conditions; light/dark cycles of 12/

12 hours. The acclimatization period was one week. During the acclimatization and the experimental period, 6 mice were put together per cage; cage maintenance and animal health monitoring was performed twice daily by veterinary nurses at the laboratory animal facility at Haukeland University Hospital. Humane endpoints were defined prior to study start. Animals observed with reduced activity level and no food or water intake for > 12 hours, with weight loss >20% or absolute weight below 14 grams would be euthanized. The mice had ad libitum access to food (maintenance diet from Scanbur, Denmark) and water. To reduce suffering and distress, cages were provided with bedding and nesting material, shelters and chewing implements. Pain, and thus need for analgesics, due to experimental conditions or procedures were not expected in this study. The experiment was conducted following the recommendations of the Federation of European Laboratory Animal Science Associations, the ARRIVE guidelines (S1 Appendix), and the Norwegian Animal Research Authority approved the protocol.

## Cuprizone administration

Mice were randomized at the age of 8 weeks into eight groups of six mice each and exposed to 0.2% (w/w) cuprizone (bis-cyclohexanone-oxaldihydrazone, Sigma-Aldrich, St. Louis, MO) added to the ordinary maintenance diet for up to 6 weeks. The control group was assigned to regular food and water. For immunohistological examination, the control group was sacrificed at week 8. Mice were sacrificed weekly during CPZ exposure, as a set group of 6 mice, and one group was sacrificed two weeks after stopping CPZ exposure (Week 8). One mouse died after three weeks of cuprizone exposure, not fulfilling the humane endpoint criteria prior to death.

## Mouse brain histopathology

Mice were euthanized by $CO_2$. The brains were perfused by intracardial injection of 4% paraformaldehyde, before they were removed and post-fixed in 4% neutral buffered formalin (NBF) for seven days; then embedded in paraffin. All analyses were performed on 5 μm coronal sections at the bregma. For immunohistochemistry, the sections were dewaxed using xylene and rehydrated in serial aqueous dilutions of ethanol before antigen retrieval in the Diva Decloaker antigen retrieval solution (DV2004LX, Biocare Medical, CA, USA) at pH 6.2; 120˚C; 15 psi for 15 minutes. The sections were incubated with primary antibodies for myelin proteolipid protein, PLP (1:1000; Mouse monoclonal; Serotec; overnight at 4˚C), for mature oligodendrocytes, neurite outgrowth inhibitor A, Nogo-A (1:1000, Rabbit polyclonal; Chemicon; 1 hour at room temperature) and ermin (1:200; Mouse monoclonal; Merck Millipore; 2 hours at room temperature). The sections were then blocked with peroxidase blocking solution (Dako, Glostrup) and visualized with EnVision+ System (Dako, Glostrup) following the manufacturer's guideline (EnVision Systems: EnVision+ Dual Link, Single Reagents; HRP. Rabbit/Mouse). The tissue sections were counterstained with hematoxylin. The sections were dehydrated before mounting permanently in dibutylphthalate polystyrene xylene (DPX). For each antibody, the omission of the primary antibody served as a negative control. Normal brain tissue from the healthy controls served as a positive control. For all antibodies used, specific immunoreactivity has previously been identified in normal brain tissue. Information on immunopositivity in normal brain tissue has been published by the producers.

## Human brain histopathology

Autopsy tissue from 26 MS autopsy cases were available for characterization of lesion type, and examined for remyelinated areas. In these, remyelinated areas were identified in tissue from 11 cases. The de-and re-myelinated MS lesion areas were selected for further study, along with brain tissue from 5 non-neurological disease controls (Table 1). The material was

**Table 1. Clinical and demographic description of MS brain autopsies and control cases.**

| Case ID | Lesion type (Number) | Ermin+ cell counts (Remyelinated/non-Remyelinated area) | MS Phenotype | Gender | Age | Disease duration | Cause of death |
|---|---|---|---|---|---|---|---|
| **1** | Cr-WM (1) T1-GM (1) | 41/19 | Progressive | M | 34 | 13 | Bronchopneumonia |
| | | 7/4 | | | | | |
| **2** | In-WM (1) | 18/13 | Progressive | F | 65 | 26 | Congestive heart failure |
| | T1-GM (1) | 6/3 | | | | | |
| | T3-GM (1) | 4/3 | | | | | |
| **3** | In-WM (1) | 36/29 | NA | M | 43 | NA | NA |
| | Ac-WM (1) | 18/11 | | | | | |
| | T3-GM (3) | 8/5,9/3,7/4 | | | | | |
| **4** | In-WM (1) | 14/10 | Progressive | M | 55 | 36 | Acute pyelonephritis with sepsis |
| **5** | In-WM (1) | 17/10 | Progressive | F | 68 | 14 | Cerebral haemorrhage |
| **6** | In-WM (1) | 26/15 | Progressive | M | 83 | NA | Pseudomembraneous colitis |
| | Ac-WM (1) | 39/22 | | | | | |
| **7** | T1-GM (1) | 5/2 | Progressive | F | 62 | 28 | Bronchopneumonia |
| | T3-GM (2) | 3/1,4/1 | | | | | |
| **8** | Cr-WM (1) | 19/10 | NA | M | 52 | 8 | Acute pyelonephritis |
| | T1-GM (1) | 6/2 | | | | | |
| **9** | Cr-WM (3) | 13/8,22/15,24/16 | Progressive | M | 43 | 7 | Bronchopneumonia |
| | T4-GM (1) | 5/1 | | | | | |
| | T2-GM (1) | 3/3 | | | | | |
| **10** | In-WM (2) | 28/19,21/16 | NA | F | 56 | 26 | Bronchopneumonia |
| | T4-GM (2) | 6/3,4/1 | | | | | |
| **11** | T3-GM (1) | 6/2 | NA | F | 46 | NA | Hyperthermia |
| **Control** | | | | | | | |
| **1** | | | | F | 37 | | Epilepsy |
| **2** | | | | M | 53 | | Diabetes mellitus |
| **3** | | | | F | 47 | | Heart disease (HSCRT) |
| **4** | | | | M | 40 | | Endocarditis |
| **5** | | | | F | 30 | | Lymphangioleiomyomatosis |

NA = Information not available, WM = White matter, GM = Gray matter, Ac = Active, Cr = Chronic active In = Inactive, T1 = Type 1, T3 = Type 3, T4 = Type 4.

obtained from the Norwegian MS Biobank (Department of Pathology, Haukeland University Hospital, Bergen, Norway). Information on the brain region studied was unfortunately not available for many of the tissue blocks studied. Pairwise comparisons were made within the individual tissue blocks, in order to correct for possible regional differences. White- and grey matter lesions, subclassified into de- and remyelinated areas, were identified by three individual investigators, in consensus. Remyelinated areas were identified by a pattern of PLP immunostaining with less dense, less organized and thinner myelin sheaths [4, 5]. All MS-lesions containing both de-and remyelinated areas was studied. In total, 14 white matter and 15 grey matter lesion areas were investigated (Table 1).

Sections from paraffin-embedded human brain autopsy tissue (5μm) were immunohistochemically stained for myelin proteolipid protein, PLP (1:1000, Rabbit polyclonal, Abcam; overnight at 4˚C), HLA-DR (1:20, Mouse monoclonal, Dako; hours at room temperature). Nogo-A (1:200, Rabbit polyclonal, Bio-Rad; hours at room temperature) and ermin (1:200, Rabbit polyclonal, Sigma-Aldrich; hours at room temperature). Processing of paraffin-

embedded tissue section for immunohistochemistry was the same as for mouse tissue sections. (PLP and HLA- DR isotype immunostainings were used to identify and classify MS lesions.

Double-labelling immunohistochemistry was performed using the EnVision™ G|2 Double stain System with two peroxidase steps, according to the manufacturer's recommended protocols (http://www.agilent.com). Tissue sections were incubated for 2 hours at room temperature with the primary antibodies followed by 30 minutes with the appropriate peroxidase-labelled secondary antibodies. Double stainings were performed for ermin (1:200, Rabbit polyclonal, Sigma-Aldrich) together with three stage-specific oligodendrocyte (OL) maturation markers: O4, O1 and Nogo-A [20, 21]. O4 (1:50, Mouse monoclonal, MyBioSource); O1 (1:50, Mouse monoclonal, Thermo Fisher Scientific Inc.) and Nogo-A (1:200, Rabbit polyclonal, Bio-Rad Laboratories). Secondary antibodies were visualized by 3,3′-Diaminobenzidine (DAB) and by reddish Alkaline Phosphatase (AP).

## Assessment of the ermin and Nogo-A positive cell population and PLP positive areas in the mouse brain

In mouse brain tissue, the density of immunopositive cell populations (ermin immunopositive and Nogo-A immunopositive cells) was determined by light microscopy (Zeiss® Axio Imager A2, Wetzlar, Germany) using an ocular morphometric grid. Observed areas were lateral corpus callosum, medial corpus callosum, supplementary motor cortex and deep grey matter in order to characterize possible regional or tissue-specific differences in immunopositivity for ermin and/or Nogo-A. Immunopositive cells were identified by brown (DAB) staining along with blue (hematoxylin) nuclear stain. Immunopositive cell density count was made in an area of 0.0625 mm$^2$. For the corpus callosum area, lateral and medial counts were averaged. Only cells with a preserved nucleus and oligodendrocyte morphology were counted. To quantify myelin content, greyscale images of tissue sections stained with PLP were thresholded, avoiding low-intensity background staining. The myelin content was expressed as the percentage of positive pixels in each PLP-stained image within the threshold values, using image processing and analysis in ImageJ (U. S. National Institutes of Health; Bethesda 2009). All sections and images were blinded with respect to group allocation for all assessors.

## Assessment of ermin+ and Nogo-A+ cell density in the human brain

Regions of interest (ROI) for counting were systematically randomized using grids. Grid-wise reference landmarks were utilized for identifying identical areas on ermin, Nogo-A and PLP immunostained tissue slides. To avoid regional bias, the comparisons were made between non-remyelinated and myelinated grey- or white matter areas having a similar spatial orientation (i.e. similar distance from the lesion edge/centre and similar tissue morphology) within the same tissue block. Ermin+ and Nogo-A+ cell densities were the nearest whole number of the average of three randomly selected areas (0.0625 mm$^2$ / area). For relatively smaller lesions, a single count was made since three randomly selected areas could not fit inside remyelinated regions.

## Statistical analysis

Mean (M) and Standard Deviation (SD) were used to express the variability in the spatio-temporal change of ermin and Nogo-A positive cell densities in CPZ model. To investigate possible differences of density of ermin and Nogo-A immunopositive cells at each sampling point, paired sample t-test was performed. Ermin+ and Nogo-A+ cell densities were measured at each sampling point; in controls and after 1, 2, 3, 4, 5 and 6 weeks of cuprizone (CPZ) exposure along with 2 weeks after the end of CPZ exposure (week 8). The ratios of the density of ermin

+ to Nogo-A+ cells were calculated for each animal for each sampling point and averaged per group. Differences of the ratio occurring at two sequential time points during the cuprizone disease course were analyzed applying the Mann-Whitney U test. For MS, Paired sample t-test was used to compare ermin+ cell density of two different parts of the same lesion. Paired sample t-test was also carried out to examine the differences in density between ermin+ and Nogo-A+ cells both in the de- and remyelinated areas of the same lesions. Independent-group t test was carried out to investigate the difference of ermin+ and Nogo-A+ cell density in MS tissue compared to that in healthy control samples. In the cuprizone samples, the sample size was less than 10, and the Shapiro-Wilk test was used to ensure the normality of data sets. Otherwise, histogram and boxplot were used to obtain an indication. A sample size of 6 mice per experimental group detects an effect size d = 0.82 with respect to the difference in NOGO-A- and Ermin-positive cell density, given power 1-β = 0.80 and error probability α = 0.05. Statistical analyses were performed using GraphPad Prism 6.0 (GraphPad Software, Inc., San Diego, CA). Differences were considered significant at p < 0.05. The correlation of the ratio of the density of ermin+ vs. Nogo-A+ cells in demyelinated and remyelinated MS-lesion areas with patient age and disease duration was calculated using the Pearson correlation coefficient.

## Results

### Ermin and Nogo-A immunopositivity and change of myelin content during and after cuprizone exposure

Myelin sheaths and the cytoplasm of cells with the morphology of oligodendrocytes were immunopositive for ermin (Fig 1), while for Nogo-A, immunopositivity was strong for oligodendrocytes, but weaker for myelin and the pattern of Nogo-A immunopositivity in the cuprizone mice was consistent with what has been reported previously [22].

Cuprizone exposure led to progressive demyelination and oligodendrocyte loss (Fig 2). White matter was represented by the corpus callosum, and by white matter tracts within deep grey matter (Fig 1). The density of PLP-immunopositive myelin sheaths decreased from week 2 to week 6 in corpus callosum and deep grey matter but, in the cerebral cortex, myelin loss was rapid and nearly complete after 1 week of cuprizone exposure (Fig 2A and 2B). Two weeks after cuprizone exposure was stopped (week 8), there was significant (p < 0.05) remyelination in the corpus callosum and deep grey matter, but not in the cerebral cortex (Fig 2A and 2B).

The density of Ermin+ cells with oligodendrocyte morphology decreased from the beginning of CPZ exposure (Fig 2C–2E and Table 2). Already at week 3, the density of Ermin+ oligodendrocytes was significantly lower in all regions studied, compared to controls (Fig 2C–2E) (p<0.001). During the period of remyelination, two weeks after withdrawal of CPZ exposure, the density of ermin+ oligodendrocytes was significantly increased in the corpus callosum (Fig 2C–2E and Table 2) (p< 0.0001).

The ratio of ermin-to-Nogo-A immunopositive cells in the controls were 1.17, 1.07 and 3.02 in the corpus callosum, the deep grey matter and the cortex, respectively. After three weeks of CPZ exposure, the ratio peaked in the corpus callosum and the deep grey matter, at 22.53 and 4.02, respectively. At this point, the density of ermin+ cells was significantly higher than that of Nogo-A+ cells in the corpus callosum and in the cortex (p < 0.02, Table 2). The sequential change of ratios of the density of ermin+ versus Nogo-A positive cells at week 3 was significant in the corpus callosum (p < 0.02). For the other areas, the density of ermin+ cells was so low at week 3 (Fig 2) that ermin+/Nogo-A+ ratios could not be determined, having some null values in the denominator. After week 3, the ratio gradually decreased throughout the cuprizone exposure phase in all areas observed (Table 2).

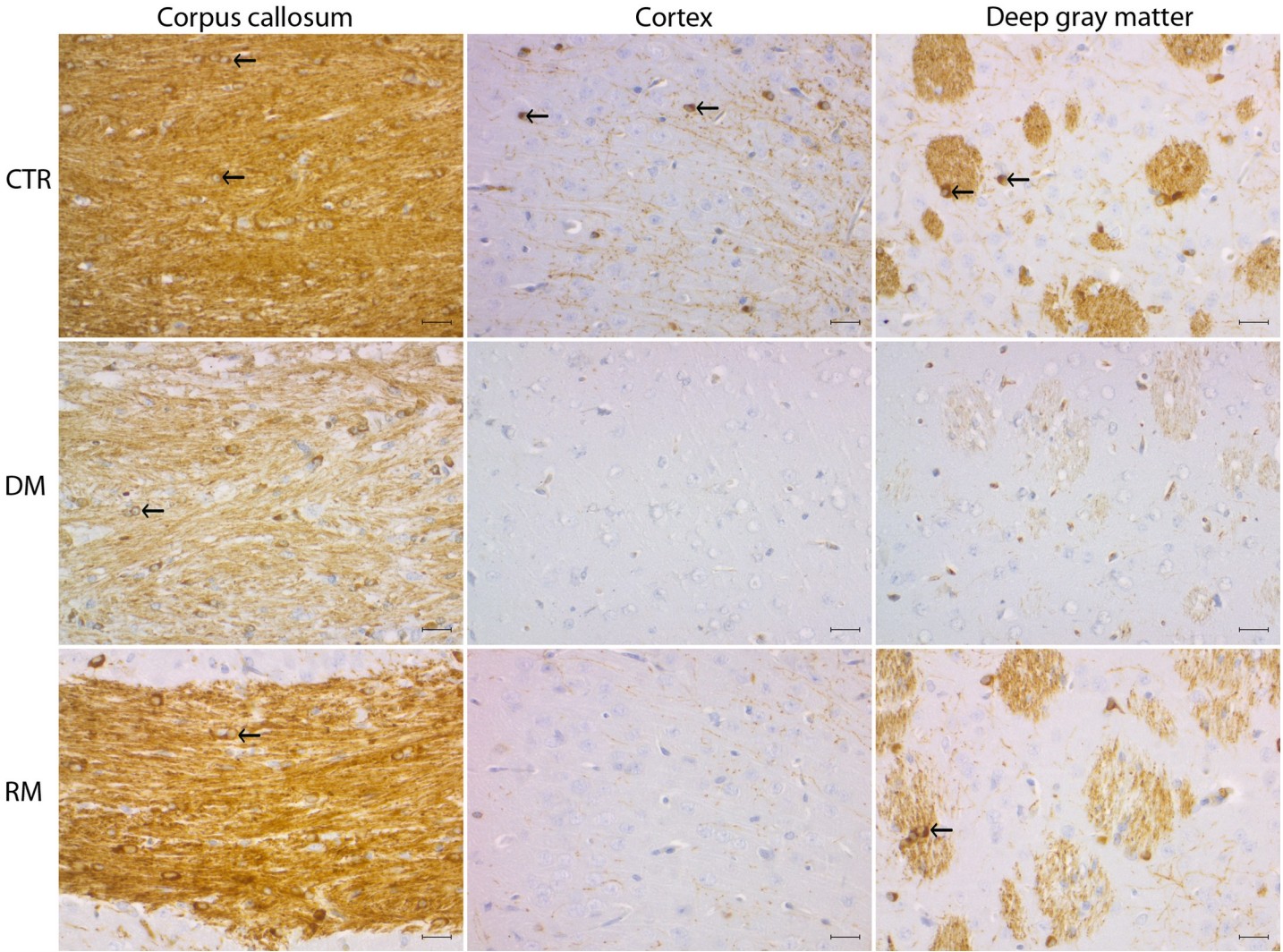

**Fig 1. Immunopositivity for ermin during de- and remyelination in the cuprizone model.** Tissue sections single labelled for ermin. First column: Corpus callosum (white matter), middle column: Cerebral cortex, last column: deep gray matter. Top row: Controls- CTR (week 8), middle row: 6 weeks of CPZ exposure (Demyelination-DM), last row: 2 weeks after the ending of CPZ exposure (Remyelination-RM). Black arrows: Cells with the morphology of oligodendrocytes, immunopositive for ermin. Myelin tracts and cells with the morphology of oligodendrocytes are immunopositive for ermin. There is extensive demyelination after 6 weeks of cuprizone exposure. At this time point, demyelination is more extensive in pure gray matter parts than in interspersed white matter in deep gray matter. Scale bar = 50 μm.

## Ermin and Nogo-A immunopositivity in MS

Adjacent tissue sections from the same MS tissue blocks were immunostained for PLP, ermin and Nogo-A. PLP immunostaining was used to define the lesion areas and the subregions of remyelination. Remyelinated areas had less dense myelin, the myelin was less organized, and myelin sheaths were thinner than in adjacent normal-appearing white- or grey matter (Figs 3A and 4A). Myelin was strongly immunopositive for ermin (Fig 5E), more weakly immunopositive for Nogo-A (Fig 5F), while cells with an oligodendrocyte morphology (both in the white and grey matter areas) were strongly immunopositive for both markers (Fig 5E and 5F). The density of ermin+ cells was significantly higher in remyelinated white matter MS lesion areas than in non-remyelinated lesion areas (p < 0.0001), or normal-appearing white matter (p < 0.001) (Fig 3B–3E). Ermin+ cell density was also increased in remyelinated grey matter

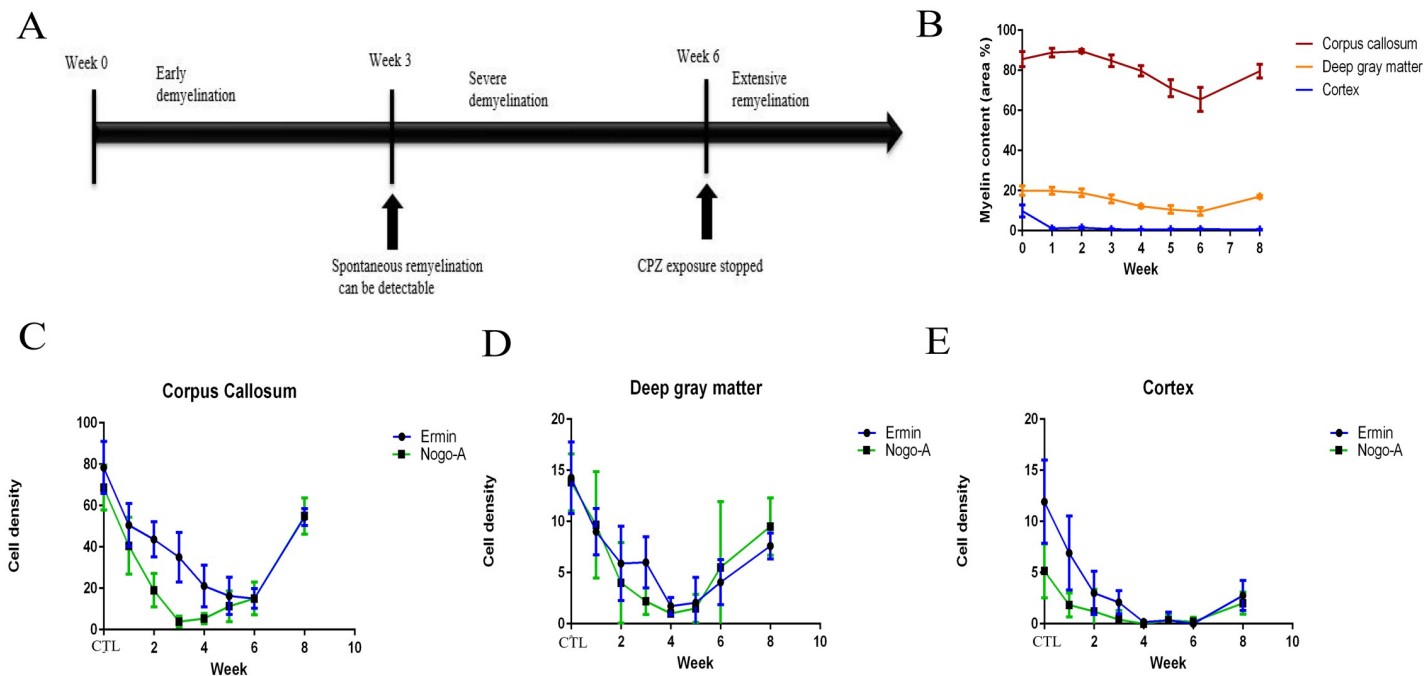

**Fig 2. Temporal dynamics of ermin and Nogo-A immunopositive cell density and myelin content over the course of de- and remyelination in the cuprizone mouse model.** (A) Schematic diagram of the course of demyelination/remyelination in the cuprizone mouse model (B) Temporal change of myelin content, measured by immunohistochemistry for proteolipid protein (PLP) (mean, SEM). (C-E) Temporal change of ermin and Nogo-A- immunopositive cell density (mean, SEM) in the corpus callosum (C), deep gray matter (D), cortex (E) from week 1 to week 8. CTL: Controls. Cuprizone exposure led to progressive demyelination and oligodendrocyte loss. The density of Nogo-A+ cells fell more rapidly than the density of ermin+ cells during the first 3 weeks (C-E). Two weeks after cuprizone exposure was stopped (week 8) remyelination was evident in corpus callosum and deep gray matter, but not in cerebral cortex (B).

lesion areas compared to non-remyelinated grey matter lesion areas (p < 0.0001), but was lower compared to normal-appearing grey matter (p < 0.01, Fig 4B–4E). For Nogo-A cell densities, a significantly higher cell density was observed in remyelinated WM compared to NAWM (p < 0.01), but not in remyelinated GM compared to NAGM (S1 Fig). Ermin+ or Nogo-A+ cell density in normal-appearing parts of white-and grey matter in MS was not significantly different from Ermin- or Nogo-A immunopositive cell density in white-and grey matter in healthy controls. The density of cells immunopositive for ermin was significantly lower in WM/GM MS-lesions compared to NAWM/NAGM (p<0.05). The density of cells immunopositive for Nogo-A was significantly lower in WM/GM lesions compared to NAWM/NAGM (p<0.05). There was no significant correlation of the density of ermin+- or Nogo-A+ cells in remyelinated areas in WM/GM with patient age (R-value Ermin -0.383/-0.58 Nogo-A -0.202/-0.021). In remyelinated lesion areas, the density of ermin+ cells was significantly higher than Nogo-A+ oligodendrocytes in white matter (p < 0.0001), but not in grey matter (p = 0.24) (Fig 5A–5F).

## Oligodendrocyte differentiation stage of ermin+ cells

MS tissue sections were double-stained for ermin and markers for stages of oligodendrocyte maturation; Nogo-A, O1 and O4. Ermin immunoreactivity was confined to only a minor subpopulation of Nogo-A+ cells with an oligodendrocyte morphology and *vice versa* (Fig 5G) and

**Table 2. Sequential change of ermin and Nogo-A positive cell density and their ratios.**

| | Week 1–6: Weeks of cuprizone exposure Week 3: Onset of remyelination | | | | | | | | | | | | | | | |
| | Control | | 1 | | 2 | | 3 (Rem) | | 4 | | 5 | | 6 | | 8 (AE) | |
| | M | SD | M | SD | M | SD | M | SD | M | SD | M | SD | M | SD | M | SD |
| *Corpus callosum* | | | | | | | | | | | | | | | | |
| Ermin | 78.4 | 12.6 | 50.6 | 10.4 | 43.6 | 8.5 | 35 | 12 | 21.2 | 10.1 | 16.3 | 9.1 | 15 | 4.8 | 52.8 | 4.8 |
| Nogo-A | 68.7 | 10.9 | 40.5 | 13.8 | 19 | 8.2 | 3.9 | 2.6 | 5.4 | 2.5 | 11.3 | 7.4 | 15.1 | 7.9 | 54.9 | 8.8 |
| Ermin: Nogo-A | 1.2 | | 1.6 | | 2.7 [a,b] | | 22.5 [a,b] | | 4.8 [a] | | 1.7 | | 1.3 | | 1 | |
| *Cortex* | | | | | | | | | | | | | | | | |
| Ermin | 11.9 | 4.1 | 6.9 | 3.6 | 3 | 2.1 | 2.1 | 1.1 | 0.2 | 0.3 | 0.3 | 0.8 | 0 | 0 | 2.8 | 1.5 |
| Nogo-A | 5.2 | 2.6 | 1.8 | 1.2 | 1.2 | 2.2 | 0.4 | 0.9 | 0 | 0 | 0.3 | 0.5 | 0.2 | 0.4 | 2 | 1.1 |
| Ermin: Nogo-A | 3.0 [a] | | 3.3 [a]* | | 0.3* | | 1.7 [a]* | | NaN | | 1* | | NaN | | 1.7 | |
| *Deep gray matter* | | | | | | | | | | | | | | | | |
| Ermin | 14.3 | 3.5 | 9 | 2.3 | 5.9 | 3.6 | 6 | 2.5 | 1.7 | 0.9 | 2.1 | 2.5 | 4.1 | 2.2 | 7.6 | 1.3 |
| Nogo-A | 13.8 | 2.8 | 9.7 | 5.2 | 4 | 3.9 | 2.2 | 1.3 | 1 | 0 | 1.5 | 1.4 | 5.5 | 6.4 | 9.5 | 2.8 |
| Ermin: Nogo-A | 1.1 | | 1.2 | | 1.3* | | 4.0 | | 1.7 | | 1.7* | | 1.1* | | 0.9 | |

Ermin+ and Nogo-A+ cell densities were measured at each sampling point; in controls, after 1, 2, 3, 4, 5 and 6 weeks of cuprizone (CPZ) exposure and 2 weeks after the end of CPZ exposure (week 8). Positive cell counts are provided as mean number per $0.0625 mm2$ (SD). The ratios of ermin+ cells compared to Nogo-A+ cells were calculated for each animal for a sampling point and averaged (M). Paired sample t-test was performed to analyze ermin:Nogo-A ratio. The change in ratio (each sampling point compared to the preceding sampling point) was analyzed by Mann-Whitney U test.

**a:** denotes an ermin:Nogo-A ratio significantly ($p<0.05$) different by paired sample t-test.

**b:** denotes a significant ($p<0.05$) change in ermin:Nogo-A ratio compared to preceding sampling point by Mann-Whitney U test.

*: denotes the presence of zero values (0) for some Nogo-A cell counts (in the deep gray matter and the cortex, Nogo-A+ cell densities were very low during cuprizone (CPZ) exposure), Nogo-A cell counts being a denominator, this excluded such areas from the calculation. **NaN:** denotes groups for which the number of sampling regions with computable ratios was too low to compute a mean ratio. **Rem**: Remyelination has been found to occur from this time point. **AE**: 2 weeks after stopping CPZ exposure

ermin protein expression was not detected in O1 and O4 positive oligodendrocyte cells, O1 and O4 are markers of oligodendrocyte progenitor cells (Fig 5H and 5I).

## Discussion

The data of this study indicate that a high relative proportion of ermin immunopositivity in white matter oligodendrocytes is associated with remyelination. Ermin immunopositivity was localized to oligodendrocytes and myelin, both in white- and grey matter parts of the mouse and human CNS. The observed specificity of ermin immunopositivity for oligodendrocytes and myelin in the CNS is consistent with what has been found previously, in both humans and in mice [13]. A higher relative density of oligodendrocytes immunopositive for ermin was found in the cerebral cortex in the 8-week old mice. This may be due to differences in the time course of myelination of the cerebral cortex compared to the corpus callosum [23].

In the cuprizone model, remyelination starts after 3 weeks of cuprizone exposure [10], and is accelerated after the exposure is ended [11]. We observed a gradual decrease of ermin+ cell density in all the observed brain areas (corpus callosum, cerebral cortex and deep grey matter), during the first 4–5 weeks of CPZ exposure. After that, the density of ermin+ cells was stable during exposure and then increased after the exposure was stopped. This pattern differed from the density of Nogo-A+ oligodendrocytes, where the density of Nogo-A+ cells fell more rapidly during the first 3 weeks; then stabilized during continued exposure and later increased. Nogo-

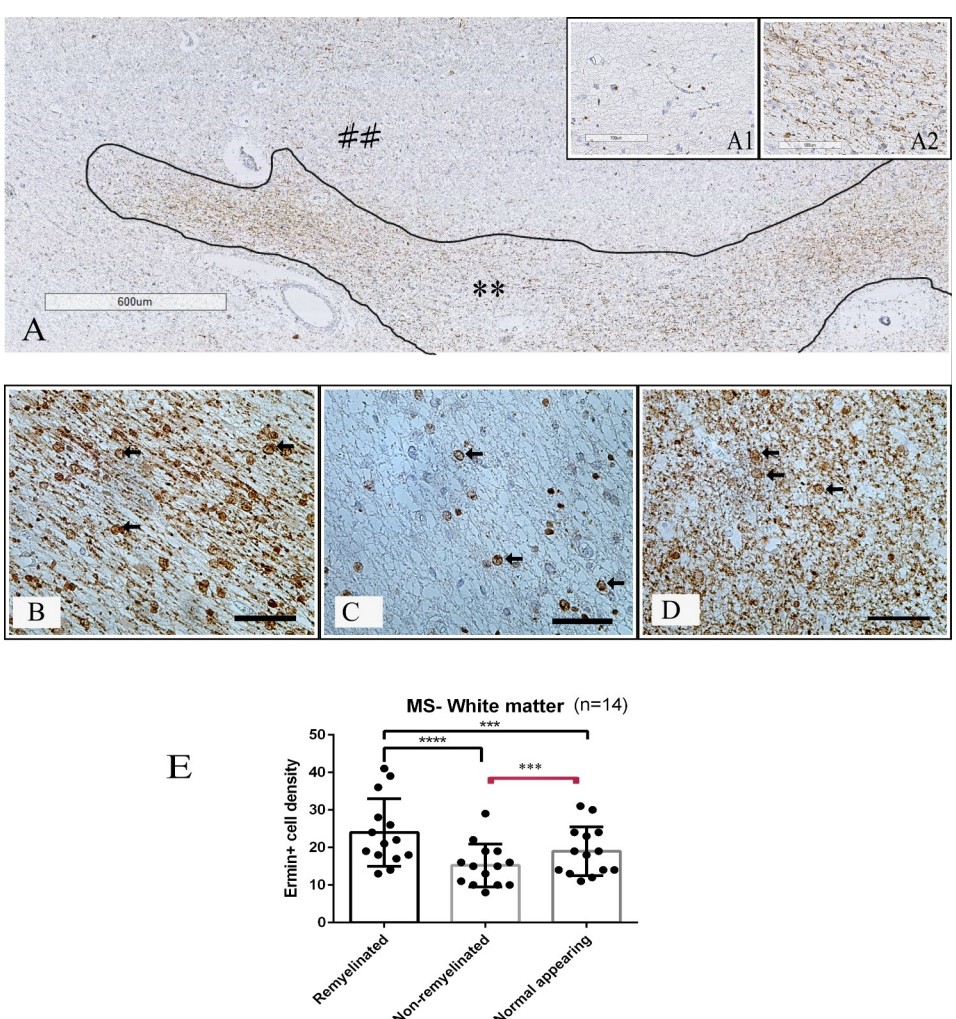

**Fig 3. Ermin positive cell density in MS white matter lesions.** (A) Myelin proteolipid protein (PLP) stained tissue section (** represents remyelinated white matter area and demyelinated white matter area is represented by ##); insertion A1 shows a demyelinated area with no remyelination, and insertion (A2) shows a partly remyelinated area. (B-D) Single-labelling immunohistochemistry for ermin in remyelinated (B) and non-remyelinated (C) lesion areas, and in normal-appearing (D) white matter tissue from MS brain. Immunopositive cells have the morphology of oligodendrocytes (arrows). (E) Density of ermin+ cells in remyelinated, non-remyelinated and normal-appearing regions in white matter. Scale bar = 50 μm, ****: p < 0.00001, ***: p < 0.0001.

A is a marker of mature oligodendrocytes in both the adult mouse and human CNS [20]. The known function of Ermin and Nogo-A differs. Ermin concentrates at the tip of F-actin-rich processes in the cytoskeleton synthesis in oligodendrocytes [14]. Ermin is likely to have a role in cytoskeleton synthesis, important for the extension and wrapping of oligodendrocyte processes around axons during myelination [13, 14]. The expression of Nogo-A may increase during myelin compaction, to trigger the collapse of F-actin in oligodendrocyte processes, a role functionally similar to Nogo-A expression in neuronal growth cones [24–27]. This may explain why the changes in ermin+- and Nogo-A+ cell populations did not completely overlap during the time course of de-and remyelination in the cuprizone model, with a relative increase in density of ermin-immuopositive cells occurring during the onset of the remyelination process.

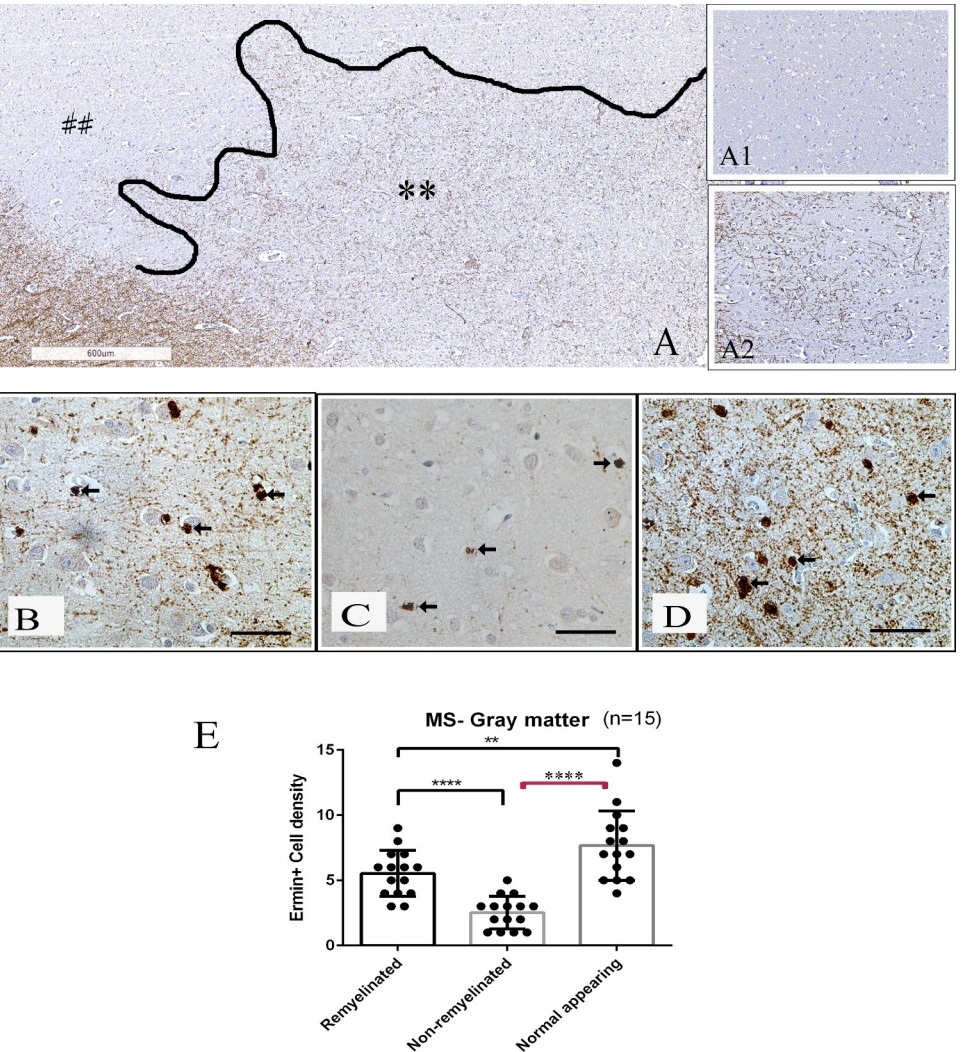

**Fig 4. Ermin positive cell density in MS gray matter lesions.** (A) Myelin proteolipid protein (PLP) stained tissue section (** represents remyelinated area, demyelinated area is represented by ##); the insertion A1 shows an area of complete demyelination in gray matter. The insertion A2 shows partial remyelination. (B–D) Single-label immunohistochemistry for ermin in regions of remyelinated (B), non-remyelinated (C) and normal-appearing (D) gray matter tissue from MS brain, immunopositive cells are marked with arrows. (E) The graph presents ermin+ cell density profile in remyelinated, non-remyelinated and normal-appearing regions in gray matter. Ermin+ cell density was increased in remyelinated gray matter lesion areas compared to non-remyelinated gray matter lesion areas ($p < 0.0001$), but was lower compared to normal-appearing gray matter ($p < 0.01$). Scale bar = 50 μm, ****: $p < 0.00001$, **: $p < 0.001$.

In a subset of oligodendrocytes, ermin- and Nogo-A immunopositivity colocalized. Ermin + cells were negative for O4, a marker for immature oligodendrocyte precursor cells and for O1, a marker for an intermediate stage of oligodendrocyte differentiation [28]. This supports ermin expression to be a marker for oligodendrocytes at a later stage of differentiation.

The increased proportion of ermin+ oligodendrocytes compared to Nogo-A+ cells in areas of remyelination may be due to an increased density of not fully mature oligodendrocytes, but it could also be partly due to a disproportionate loss of mature oligodendrocytes in these areas, as remyelination in MS and in CPZ occurs simultaneously with myelin and oligodendrocyte loss [10].

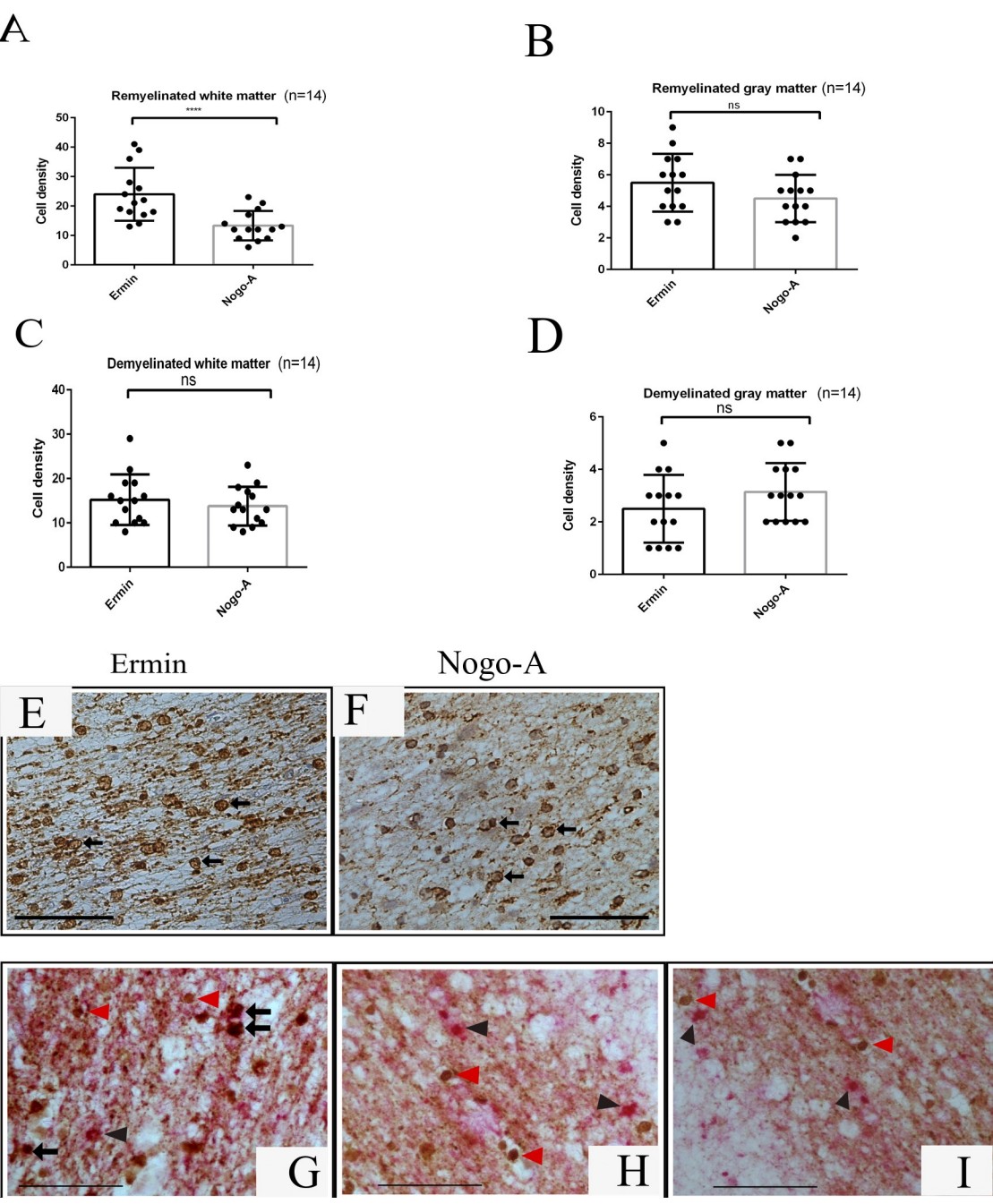

**Fig 5. Comparison of ermin and Nogo-A expression in de- and remyelinated MS lesion areas.** Density of ermin+ and Nogo-A + cells in remyelinated white matter (A) and gray matter (B) MS lesion areas, and demyelinated white matter (C) and gray matter (D) MS lesion areas. A remyelinated white matter MS lesion area single-label immunostained for ermin (E) and Nogo-A (F). White matter remyelinated MS lesion areas double-label immunostained for ermin (DAB, brown) (G-I) and Nogo-A (AP, red-pink) (G), O1 (AP, red-pink) (H), and O4 (AP, red-pink) (I). Arrows: Double-immunopositive cells. Red arrowhead: Cells immunopositive for ermin, Black arrowhead: Nogo-A (A); O1 (H) and O4 (I). A small proportion of oligodendrocytes were double immunopositive for ermin and Nogo-A (G, arrows). Ermin+ cells were not immunopositve for O1 or O4. Scale bar = 50 μm, ****: p < 0.00001, ns: Nonsignificant (p > 0.05).

The relative loss of myelin was more rapid and extensive in CPZ grey matter than in white matter tracts, consistent with what has been described previously [22]. This was evident in the cerebral cortex and in deep grey matter, where demyelination was extensive in pure grey matter parts during cuprizone exposure, but not in interspersed white matter fibre tracts after 6 weeks of cuprizone exposure. Less differentiated oligodendrocytes may be more sensitive to metabolic injury than more mature cells [29]. This difference in the time course of toxic demyelination in grey-and white matter may thus be due to possible differences in the stage of differentiation of oligodendrocytes in grey and white matter in mice at age 8–14 weeks, compared to earlier time points.

Although the density of ermin+—and Nogo-A+ -oligodendrocytes increased both in white-and grey matter areas after ending cuprizone exposure, remyelination was not detected in cerebral cortex. After the end of CPZ exposure remyelination in the cerebral cortex has previously been found to occur at a slower rate than in white matter [10]. This difference may be due to location-dependent differences in oligodendrocyte phenotype, but may also be due to other factors, such as a difference in the extent of inflammation [30, 31].

The level of ermin expression in the CNS may be correlated with ermin levels in CSF and/or in blood. A recent study has reported downregulation of the ermin gene (mRNA) in blood from RRMS patients [32]. The possible correlation of the level of ermin expression in CSF and/or blood with the extent of remyelination needs to be investigated further.

The interpretation of the data presented in this study has several important limitations.

The density of ermin-immunopositive cells was not studied in relation to markers of lesion age and inflammatory activity, as there is no consensus on markers of lesion age and activity in MS grey matter lesions. The human material was mainly composed of patients with late stages of multiple sclerosis. Mechanisms of remyelination may differ in early vs. late stages of the disease. In the cuprizone model, the expression pattern of ermin was compared to Nogo-A only, because data on the function of ermin and Nogo-A indicate these proteins may be markers of adjacent stages in myelin formation. The antibodies used as markers for oligodendrocytes/oligodendrocyte precursors may have differences in sensitivity, however, this is not likely to explain the changes observed in the proportion of ermin-to Nogo-A immunopositive cells during the period of de-and remyelination in the cuprizone mice. Additional studies on other oligodendrocyte maturation markers would help to characterize the ermin+-cell population further, including the timing of ermin expression during remyelination. Ermin expression pattern may differ in different brain regions, but the spatial brain location of all MS-brain autopsy tissue studied could not be accurately determined. Only lesions containing myelinated and non-remyelinated parts were studied, in order to avoid confounding due to regional differences. Distinguishing remyelination from active demyelination is a challenge to both this and previous studies of post-mortem tissue. Areas of remyelination in chronic MS has been found to be characterized by focal or lesion border areas of thin, irregular myelin sheaths. In these areas, few microglia and macrophages are generally found, making it unlikely to represent active demyelination in initial lesions [4–7]. Very similar criteria for light microscopy has been confirmed by electron microscopy [33].

## Conclusion

Several lines of evidence indicate that remyelination may confer neuroprotection, with axonal preservation in multiple sclerosis lesions [7]. Better knowledge of the mechanisms of remyelination in MS may provide therapeutic targets to enhance remyelination, and provide long term neuroprotection. The findings of this study suggest that remyelination is associated with a relative increase in the density of ermin-positive oligodendrocytes. The data strengthen the

hypothesis that ermin expression may be important for the early stages of myelin formation, prior to myelin compaction.

## Supporting information

**S1 Fig.** Nogo-A+ cell density in white matter **(A)** and gray matter **(B)** in MS-brain. Remyelinated, non-remyelinated and normal-appearing areas are compared. ns: Nonsignificant (p > 0.05).
(TIF)

**S1 Appendix. The ARRIVE guidelines checklist.**
(PDF)

## Author Contributions

**Conceptualization:** Stig Wergeland, Eystein Oveland, Lars Bø.

**Data curation:** Intakhar Ahmad, Eystein Oveland, Lars Bø.

**Formal analysis:** Intakhar Ahmad, Lars Bø.

**Investigation:** Intakhar Ahmad, Stig Wergeland, Eystein Oveland, Lars Bø.

**Methodology:** Intakhar Ahmad, Stig Wergeland, Eystein Oveland, Lars Bø.

**Project administration:** Stig Wergeland, Eystein Oveland, Lars Bø.

**Resources:** Eystein Oveland.

**Supervision:** Stig Wergeland, Eystein Oveland, Lars Bø.

**Validation:** Stig Wergeland, Eystein Oveland, Lars Bø.

**Writing – original draft:** Intakhar Ahmad, Stig Wergeland, Eystein Oveland, Lars Bø.

**Writing – review & editing:** Intakhar Ahmad, Stig Wergeland, Eystein Oveland, Lars Bø.

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
