## [Decision Letter · Decision Letter 0]

13 Apr 2021

PONE-D-21-07935

Ermin is a potential marker of recent or ongoing remyelination

PLOS ONE

Dear Dr. Bø,

Thank you for submitting your interesting manuscript to PLOS ONE. After careful consideration, we feel that it has merit but does not fully meet PLOS ONE’s publication criteria as it currently stands. Therefore, we invite you to submit a revised version of the manuscript that addresses the points raised during the review process.

We look forward to receiving your revised manuscript.

Kind regards,

Catherine FAIVRE-SARRAILH

Academic Editor

PLOS ONE

Journal Requirements:

Reviewers' comments:

Reviewer's Responses to Questions

**Comments to the Author**

1. Is the manuscript technically sound, and do the data support the conclusions?

Reviewer #1: Yes

Reviewer #2: No

2. Has the statistical analysis been performed appropriately and rigorously? 

Reviewer #1: Yes

Reviewer #2: Yes

3. Have the authors made all data underlying the findings in their manuscript fully available?

Reviewer #1: Yes

Reviewer #2: Yes

4. Is the manuscript presented in an intelligible fashion and written in standard English?

Reviewer #1: Yes

Reviewer #2: Yes

5. Review Comments to the Author

Reviewer #1: This is a well-conducted study investigating whether Ermin could be used as a biomarker to evaluate the extent of remyelination in MS lesions. Ermin is an actin-binding protein likely implicated in the extension and wrapping of oligodendrocyte processes around axons and it is expressed in differentiated oligodendrocytes earlier than Nogo-A. I have only minor comments.

Please cite the recent paper in Glia 2020 Wang S et al about the phenotype of the ermin KO mice. It is reported that these mice exhibit an aberrant myelin architecture in aged mice with the splitting of myelin layers and that they display an increased sensitivity to cuprizone demyelination.

Is there any correlation between the density of ermin or nogo-A-positive cells in remyelinated lesions with the age of the patients? Maybe this would require a larger cohort.

Could you comment on the relatively high density of ermin+ by comparison with nogo-A+ cells (more than twice) specifically in the gray matter of the cortex? could it be dependent on the layers of the cortex, on the age (8 weeks) of the mice?

Minor typos

Lane 129 rabbit

Fig 5B correct gary instead of gray

Lane 279 O1 and O4 are markers of oligodendrocyte progenitor cells

Reviewer #2: In the manuscript “Ermin is a potential marker of recent or ongoing remyelination” Ahmad et al. perform a comparative histopathological analysis of Nogo-A and Ermin-expressing cells in human post mortem MS tissue and in different timepoints during and after cuprizone administration in rodents. They note a significant difference in the density of Ermin+ and Nogo-A+ cells in the first weeks of cuprizone administration and in remyelinated white matter in the human post-mortem tissue. Authors then conclude that Ermin is a rather earlier marker than Nogo-A and characteristic of ongoing remyelination.

Although the expression profile of Ermin+ cells in the human tissue and the comparison with other markers of the oligodendrocyte lineage would be of potential interest, complementing works done in the past (such as on Nogo-A , DOI:10.1097/01.jnen.0000248559.83573.71) the results the authors present don’t really support the conclusions that the authors finally draw. At the moment, there is no strong evidence that Ermin is a marker of recent or ongoing remyelination.

Ermin and Nogo-A are both considered markers of late oligodendrocyte differentiation and maturation and were both located in the non-compact myelin parts of the myelin sheath as assessed with immune gold EM ( DOI: https://doi.org/10.1523/JNEUROSCI.22-09-03553.2002, DOI: https://doi.org/10.1523/JNEUROSCI.4317-05.2006 ). Nogo-A colocalises also with CC1 another marker of mature oligodendrocytes that includes both myelinating and pre-myelinating oligodendrocytes (DOI:10.1097/01.jnen.0000248559.83573.71).

Ermin was shown to be expressed rather late by the myelinating oligodendrocyte even slightly after MBP expression in vitro and after CNPase in vivo (DOI: https://doi.org/10.1523/JNEUROSCI.4317-05.2006, DOI: 10.1002/gli a.23838, https://doi.org/10.1073/pnas.0500952102). These studies favour more the earlier expression of Nogo-A rather than that of Ermin as the authors claim in the present manuscript. Although both markers can be expressed by mature oligodendrocytes, in the present manuscript there is no indication that the expressing cells are mature cells that hadn’t yet undergone complete demyelination at least in the case of the cuprizone data. The difference in the number of Nogo-A and Ermin expressing cells during the cuprizone demyelination period is quite striking but it does not preclude differences in the sensitivity of the antibodies used.

My worry is that since Ermin antibody stains also the myelin sheaths, it makes the precise quantification of oligodendrocytic densities difficult as in the case of other myelin markers (i.e.PLP,MBP,MOG) in areas where myelin is dense. In Fig1A the signal is uniform and dense and according to the graph in Fig2B the myelin content is not reduced until after week 4 in cuprizone treated animals for corpus callosum. During this period (1-3 weeks of cuprizone) there is a lot of myelin debris around engulfed by microglia so the difference in the Ermin+ versus the Nogo-A+ numbers can indicate dying cells rather than remyelinating ones at least in the corpus callosum.

In the cortex (Figure 2E) the density of control Ermin+ oligodendrocytes is nearly double than that of Nogo-A. Also, the image in Fig1I does not agree with the myelin content in Fig2B.

In none of the areas analysed there is a difference in the populations during remyelination after cuprizone removal and it is not clear how they detect spontaneous remyelination.

Authors need to reconsider their outcomes and provide additional quantifications in combination with other mature (such as CC1, myrf) or general lineage markers like Olig2 to strengthen their initial claims. They also need to show that these Ermin expressing cells are not just dying cells that haven’t yet been cleared by microglia. Perhaps an in-situ hybridization for Ermin might be more conclusive in areas of dense myelin than the antibody.

The expression pattern of Ermin and Nogo-A cells in human lesions can be potentially interesting. In the white matter areas of sparse myelination the numbers between demyelinated areas and NAWM are similar which is impressive and it would be interesting to see if the numbers are different to those of control tissue white matter. Authors need also to comment on why they believe there is no difference between NAWM and demyelinated lesion Ermin numbers and if they observe this difference with Nogo-A cell densities in these areas as well.

An additional quantification with a different cytoplasmic marker of mature oligodendrocytes will show that indeed there is high presence of mature oligos in the area that express Ermin but not Nogo-A. Authors also need to show that there is no inflammation in the areas of remyelination and also show if there is a correlation with age and disease duration if possible.

6. PLOS authors have the option to publish the peer review history of their article (what does this mean?). If published, this will include your full peer review and any attached files.

Reviewer #1: No

Reviewer #2: No

---

## [Author Response · Author response to Decision Letter 0]

21 Jul 2021

Reviewer #1

Please cite the recent paper in Glia 2020 Wang S et al about the phenotype of the ermin KO mice. It is reported that these mice exhibit an aberrant myelin architecture in aged mice with the splitting of myelin layers and that they display an increased sensitivity to cuprizone demyelination.

This paper is now cited in the manuscript (Reference #19).

Is there any correlation between the density of ermin or nogo-A-positive cells in remyelinated lesions with the age of the patients? Maybe this would require a larger cohort.

To study such differences we would have needed a larger cohort. In our data set we could not find any significant correlation (now mentioned in lines 284:286)

Could you comment on the relatively high density of ermin+ by comparison with nogo-A+ cells (more than twice) specifically in the gray matter of the cortex? could it be dependent on the layers of the cortex, on the age (8 weeks) of the mice?

The differences in the density of ermin immunopositive cells may be due to a later myelination in mouse cortex vs. corpus callosum, as has been described previously. This information has now been added to the discussion, and a reference has been added (Reference #23).

Fig 5B correct gary instead of gray

This has been corrected.

Lane 279 O1 and O4 are markers of oligodendrocyte progenitor cells

This has now been added, line number: 296

Reviewer #2: 

The authors then conclude that Ermin is a rather earlier marker than Nogo-A and characteristic of ongoing remyelination. 

Although the expression profile of Ermin+ cells in the human tissue and the comparison with other markers of the oligodendrocyte lineage would be of potential interest, complementing works done in the past (such as on Nogo-A , DOI:10.1097/01.jnen.0000248559.83573.71) the results the authors present don’t really support the conclusions that the authors finally draw. At the moment, there is no strong evidence that Ermin is a marker of recent or ongoing remyelination. Ermin and Nogo-A are both considered markers of late oligodendrocyte differentiation and maturation and were both located in the non-compact myelin parts of the myelin sheath as assessed with immune gold EM ( DOI: https://doi.org/10.1523/JNEUROSCI.22-09-03553.2002, DOI: https://doi.org/10.1523/JNEUROSCI.4317-05.2006 ).

The manuscript does not claim that ermin alone is a marker of remyelination, but that the relative proportion of oligodendrocytes immunopositive for ermin is associated with remyelination. This is now reflected by a change of title for the manuscript, and by a change in the discussion, line 321-324.

Nogo-A colocalises also with CC1, another marker of mature oligodendrocytes that includes both myelinating and pre-myelinating oligodendrocytes (DOI:10.1097/01.jnen.0000248559.83573.71).

Ermin was shown to be expressed rather late by the myelinating oligodendrocyte even slightly after MBP expression in vitro and after CNPase in vivo (DOI: https://doi.org/10.1523/JNEUROSCI.4317-05.2006, DOI: 10.1002/gli a.23838, https://doi.org/10.1073/pnas.0500952102). These studies favour more the earlier expression of Nogo-A rather than that of Ermin as the authors claim in the present manuscript. Although both markers can be expressed by mature oligodendrocytes, in the present manuscript there is no indication that the expressing cells are mature cells that hadn’t yet undergone complete demyelination at least in the case of the cuprizone data. The difference in the number of Nogo-A and Ermin expressing cells during the cuprizone demyelination period is quite striking but it does not preclude differences in the sensitivity of the antibodies used. 

The antibodies used as markers for oligodendrocytes/oligodendrocyte precursors may have differences in sensitivity, however this not likely to explain the changes observed in the proportion of ermin- to Nogo-A immunopositive cells during the period of de-and remyelination in the cuprizone mice. This has now been included in the discussion, line 361-364. 

There are, as mentioned by the reviewer, other (and earlier) markers of oligodendrocyte maturation. The relation of these other markers, and of the relative proportion of these markers to the occurrence of remyelination has not been established. This was beyond the scope of the present study. 

My worry is that since Ermin antibody stains also the myelin sheaths, it makes the precise quantification of oligodendrocytic densities difficult as in the case of other myelin markers (i.e.PLP,MBP,MOG) in areas where myelin is dense. In Fig1A the signal is uniform and dense and according to the graph in Fig2B the myelin content is not reduced until after week 4 in cuprizone treated animals for corpus callosum. During this period (1-3 weeks of cuprizone) there is a lot of myelin debris around engulfed by microglia so the difference in the Ermin+ versus the Nogo-A+ numbers can indicate dying cells rather than remyelinating ones at least in the corpus callosum. 

Only cells with a preserved nucleus and oligodendrocyte morphology were counted, thus not dying cells or myelin debris. This has now been included in Materials and Methods section of the manuscript, line 188-189.

In the cortex (Figure 2E) the density of control Ermin+ oligodendrocytes is nearly double than that of Nogo-A. Also, the image in Fig1I does not agree with the myelin content in Fig2B.

We have revised the figure number 1, to be in better accord with the findings. 

In none of the areas analysed there is a difference in the populations during remyelination after cuprizone removal and it is not clear how they detect spontaneous remyelination. 

Remyelination has previously been shown to occur from week 3 in the cuprizone model (reference #10). At that time point we found a significant change of the ratio of density of ermin+ vs. Nogo-A+ cells. We have made a change in the results text (line 258) to make this more clear. 

Later during the disease remyelination is detected by an increase in the proportion of myelinated area.

Authors need to reconsider their outcomes and provide additional quantifications in combination with other mature (such as CC1, myrf) or general lineage markers like Olig2 to strengthen their initial claims. They also need to show that these Ermin expressing cells are not just dying cells that haven’t yet been cleared by microglia. Perhaps an in-situ hybridization for Ermin might be more conclusive in areas of dense myelin than the antibody.

Only cells with a preserved nucleus and oligodendrocyte morphology were counted, thus not dying cells or myelin debris. This has now been included in Materials and Methods section of the manuscript, line 188-189. 

There are, as mentioned by the reviewer, other (and earlier) markers of oligodendrocyte maturation. The relation of these other markers, and of the relative proportion of these markers to the occurrence of remyelination has not been established. This was beyond the scope of the present study. 

The expression pattern of Ermin and Nogo-A cells in human lesions can be potentially interesting. In the white matter areas of sparse myelination the numbers between demyelinated areas and NAWM are similar, which is impressive and it would be interesting to see if the numbers are different to those of control tissue white matter. Authors need also to comment on why they believe there is no difference between NAWM and demyelinated lesion Ermin numbers and if they observe this difference with Nogo-A cell densities in these areas as well. 

A higher density of ermin+ cells was found in remyelinated WM compared to NAWM as described in results, line 272-274, and in Figure 3E. For Nogo-A cell densities, a significantly higher cell density was observed in remyelinated GM compared to NAGM, but not in remyelinated WM compared to NAWM (now provided as Suppl. Fig 1). The density of cells immunopositive for ermin and Nogo-A was significantly lower in MS-lesion areas than in NAWM/NAGM respectively. This is now included in the results section, line 281-284.

The density of ermin-and Nogo-A immunopositive cells in NAWM and NAGM was not significantly different from the density of ermin-or Nogo-A immunopositive cells in white matter or grey matter in controls. This has now been added to the results section, line 278-281. 

An additional quantification with a different cytoplasmic marker of mature oligodendrocytes will show that indeed there is high presence of mature oligos in the area that express Ermin but not Nogo-A. Authors also need to show that there is no inflammation in the areas of remyelination and also show if there is a correlation with age and disease duration if possible

Remyelination in MS occurs in areas both with-and without extensive inflammation. It was not the scope of this work to investigate the expression of ermin to stages of lesion inflammation. Such a classification would not be applicable to the MS lesions entirely within grey matter, as these show little or no lymphocyte or monocyte infiltration.

To address the correlation with age and disease duration we have added an extra column in Table 1

In statistics part: line number 228-230

We have added a supplementary figure (Suppl. Fig 1) to show Nogo-A+ cell density in different MS tissue types and different parts of lesion. Line number: 276-278

---

## [Decision Letter · Decision Letter 1]

2 Aug 2021

A higher proportion of ermin-immunopositive oligodendrocytes in areas of remyelination

PONE-D-21-07935R1

Dear Dr. Bø,

We’re pleased to inform you that your manuscript has been judged scientifically suitable for publication and will be formally accepted for publication once it meets all outstanding technical requirements.

Kind regards,

Catherine FAIVRE-SARRAILH

Academic Editor

PLOS ONE

Additional Editor Comments (optional):

Reviewers' comments:

Reviewer's Responses to Questions

**Comments to the Author**

1. If the authors have adequately addressed your comments raised in a previous round of review and you feel that this manuscript is now acceptable for publication, you may indicate that here to bypass the “Comments to the Author” section, enter your conflict of interest statement in the “Confidential to Editor” section, and submit your "Accept" recommendation.

Reviewer #1: All comments have been addressed

Reviewer #2: All comments have been addressed

2. Is the manuscript technically sound, and do the data support the conclusions?

Reviewer #1: Yes

Reviewer #2: Yes

3. Has the statistical analysis been performed appropriately and rigorously? 

Reviewer #1: Yes

Reviewer #2: Yes

4. Have the authors made all data underlying the findings in their manuscript fully available?

Reviewer #1: Yes

Reviewer #2: Yes

5. Is the manuscript presented in an intelligible fashion and written in standard English?

Reviewer #1: Yes

Reviewer #2: Yes

6. Review Comments to the Author

Reviewer #1: (No Response)

Reviewer #2: The authors have succesfully addressed my comments on the earliest version of this manuscript.

The current version of the manuscript is acceptable for publication.

7. PLOS authors have the option to publish the peer review history of their article (what does this mean?). If published, this will include your full peer review and any attached files.

Reviewer #1: No

Reviewer #2: No

---

## [Editor Report · Acceptance letter]

18 Aug 2021

PONE-D-21-07935R1 

A higher proportion of ermin-immunopositive oligodendrocytes in areas of remyelination 

Dear Dr. Bø:

I'm pleased to inform you that your manuscript has been deemed suitable for publication in PLOS ONE. Congratulations! Your manuscript is now with our production department. 

Kind regards, 

on behalf of

Dr. Catherine FAIVRE-SARRAILH 

Academic Editor

PLOS ONE